# Aspect Ratio Effects on the Aerodynamic Performance of a Biomimetic Hummingbird Wing in Flapping

**DOI:** 10.3390/biomimetics8020216

**Published:** 2023-05-23

**Authors:** Yilong Min, Gengyao Zhao, Dingyi Pan, Xueming Shao

**Affiliations:** State Key Laboratory of Fluid Power and Mechatronic Systems, Department of Engineering Mechanics, Zhejiang University, Hangzhou 310027, China

**Keywords:** biomimetic hummingbird wing, aspect ratio, computational fluid dynamics simulation, flapping wing, aerodynamic characteristic

## Abstract

Hummingbirds are flapping winged creatures with unique flight mechanisms. Their flight pattern is more similar to insects than other birds. Because their flight pattern provides a large lift force at a very small scale, hummingbirds can remain hovering while flapping. This feature is of high research value. In order to understand the high-lift mechanism of hummingbirds’ wings, in this study a kinematic model is established based on hummingbirds’ hovering and flapping process, and wing models imitating the wing of a hummingbird are designed with different aspect ratios. Therefore, with the help of computational fluid dynamics methods, the effect of aspect ratio changes on the aerodynamic characteristics of hummingbirds’ hovering and flapping are explored in this study. Through two different quantitative analysis methods, the results of lift coefficient and drag coefficient show completely opposite trends. Therefore, lift–drag ratio is introduced to better evaluate aerodynamic characteristics under different aspect ratios, and it is found that the lift–drag ratio reaches a higher value when AR = 4. A similar conclusion is also reached following research on the power factor, which shows that the biomimetic hummingbird wing with AR = 4 has better aerodynamic characteristics. Furthermore, the study of the pressure nephogram and vortices diagram in the flapping process are examined, leading to elucidation of the effect of aspect ratio on the flow field around hummingbirds’ wings and how these effects ultimately lead to changes in the aerodynamic characteristics of the birds’ wings.

## 1. Introduction

Biological flight in nature comprises a large number of aerodynamic mechanisms and has long been helpful and served as an inspiration for aircraft design. In recent years, the development of micro aerial vehicles (MAVs) has been growing rapidly, and some new problems are being encountered. The small size and light weight of these aircraft lead to a wing design characterized by a short wingspan while flying at a small Reynolds number, and it has therefore been a big challenge for MAVs to provide enough lift and thrust. For this reason, many aerodynamicists have turned their interest to insect aerodynamics and are developing insect-like or insect-inspired flapping wing micro aerial vehicles (FMAVs). Compared with birds, insects fly at much lower Re numbers in a separated flow environment [1,2], and their unique unsteady flight patterns contain many mechanisms of providing high lift force, including clap and fling [3,4,5], delayed stall [1,6,7,8,9], rapid pitch rotation [1] and wake capture [10,11]. Other insects, such as drosophila [12], locust [13], hoverflies [14], dragonflies [15,16] and mosquitoes [17], also use different kinds of aerodynamic characteristics. Rather than steady-state aerodynamics, these high-lift mechanisms allow insects to generate higher lift at smaller sizes.

The high-lift mechanism of natural organisms provides ideas for the research of aircraft, and many studies on flapping wing have accordingly been carried out [18,19,20]. On this basis, many FMAVs have been designed based on various unsteady mechanisms of biological flight in nature, including Delfly [21], KUBeet-S from Konkuk University [22], the FMAV designed at Harvard University [23], and Nano Hummingbird from Aerovironment [24]. Of these FMAVs, Nano Hummingbird and KuBeet-S refer more or less to the flight mechanism of hummingbird. A hummingbird is a creature between an insect and a bird; it has a higher weight than the average insect and its wingspan ranges between that of an insect and a bird, from 35 to 152 mm among species [25]. One of the remarkable features of hummingbirds is that they can hover steadily for a long period of time while having high maneuverability [26]. During their flight, the majority of the lift of hummingbirds comes from the leading-edge vortices (LEVs) provided by their wings [27,28]. While hovering, hummingbirds flap and revolve theirs wings in different directions at the same time, and it is generally assumed that these flight patterns of hummingbirds are very similar to those of insects [29], but their mechanisms operate at completely different Reynolds (Re) numbers. Due to their larger size and wingspan, hummingbirds usually fly at a Reynolds number regime of 5000–30,000 [30], while insects operate at much lower Reynolds number, ranging from 10 to 5000 [31]. Moreover, hummingbirds usually flap their wings at a frequency of tens of Hz when flying and hovering, much less than the frequency of insects [32]. These factors lead to the fact that even if the flight mechanism of hummingbirds is similar to insects, there may still be some differences between their aerodynamics and flight characteristics.

Studies have shown that hummingbirds have a high stroke amplitude in hovering flight [33,34] and they can operate their wings to extremely high angles of attack without stalling and create attached LEVs on the wing to provide additional lift [35]. While in midstroke of wing flapping, the stroke of the wings is reversed, and the acceleration is approximately zero, which can be approximated to constant wing rotation motion [1,26,36]. At the same time, this quasi-steady flapping phase plays an important role during downstroke, and at midstroke an attached LEV elevates lift, providing up to 75% of the lift [35,36,37,38]. This makes hummingbird wing flapping a suitable target for quasi-steady state analysis. The design of the wing largely determines the lift, drag and power performance of the hummingbird during flapping and hovering. Aspect ratio is one of the important parameters to describe the shape of wings, it is a dimensionless number defined as the ratio of wingspan to chord length. Aspect ratio significantly affects the flight performance of aircraft to a large extent, especially under the condition of high Reynolds number, and it can be the most important parameter for optimizing the performance of aircraft [39] and helicopter [40]. Despite the wide difference in wingspan, hummingbirds have roughly similar aspect ratio values, ranging from 3 to 4.5 [25,41]. This aspect ratio is much smaller than that of common aircraft. The low aspect ratio makes the effect of rotational acceleration on the boundary layer more significant than that of inertial force, which is conducive to the attachment of LEV on the wing [42]. However, these studies have not considered the effect of changes in aspect ratio on the power efficiency, in which Kruyt’s [26] study has confirmed that under the same changing conditions, aspect ratio will have an impact on the power consumption. In addition, most of the Reynolds numbers selected in the above studies are small, and at high Re numbers, the influence of inertial forces on the flow field is more significant, which makes the influence of the aspect ratio on the aerodynamic characteristics of hummingbird hovering flight more significant.

In this study, a flapping kinematic model of revolving wing is designed based on the flight mode of hummingbirds in reality, and a set of numerical models of bionic wings is created by referring to the shape of actual hummingbirds. According to the actual range of flight Reynolds number of hummingbirds, a relatively larger Reynolds number is selected in order to make the effects of aspect ratio more significant. Moreover, a series of hummingbird-like wing models with different aspect ratios are modified under the premise of maintaining a constant Re number, and the aerodynamic characteristics and flow field of wings through the computational fluid dynamics (CFD) simulation methods are obtained at different aspect ratios. After a series of studies on wing kinematics, force generation, and vortex analysis, a series of important aerodynamic and kinematic characteristics of hummingbirds are revealed, which can be used to optimize the design of hummingbird-like MAVs.

## 2. Materials and Methods

In order to explore the effects of aspect ratio on the aerodynamic characteristics of hummingbirds’ flapping wings, in the study a series of wing models with different aspect ratios (ARs) are created, and a hummingbird-like flapping motion mode is established. Numerical simulation is carried out based on the CFD method to explore the aerodynamic characteristics under flapping wings with different ARs.

### 2.1. Wing Model Design

Many studies have been conducted to analyze the biological statistics of the wingspan and aspect ratio [43,44] of hummingbird wings. According to statistical studies, the wingspan of hummingbirds varies widely between species, while the aspect ratio is roughly the same, in a general range of 3 to 4.5 [25]. Kryut [26] further analyzed and compared several photos of various kinds of dried hummingbird wings. From the comparison results of the photos, it can be seen that the shape of all hummingbirds’ wings is basically the same. Near the root of the wings, wing shape can be approximated to a trapezoid or small rectangle, while in the part beyond the half-span, as the wings gradually develop to the wing tip, the trailing edge of the wings gradually presents an arc contraction transition until it meets the leading edge at the wing tip.

The shape of the wing in this study is finally designed based on former studies. Figure 1 shows a sketch of the basic numerical model of the hummingbird-like MAV and its wings. The design idea refers to the above studies results on hummingbird wings. The trailing edge of the wing exhibits a curved edge that shrinks toward the leading edge and eventually meets the leading edge at the wing tip. Considering that the actual wing of a hummingbird mainly provides lift through soft feathers and veins, in the wing model designed by us, the part of the area near the leading edge of the wing root is removed.

After referring to the values of biological hummingbirds, a larger Reynolds number of 27,000 is selected to remain unchanged within the range of hummingbirds’ Reynolds number (5000–30,000 [30]), under the principle of making the effect of aspect ratio more significant. The dimensionless Re number is calculated by Re=uL/ν, where the characteristic length *L* is the chord length of the wing, the characteristic speed *u* used to calculate the Reynolds number is expressed as the average speed at the wing tip over a period as u=2πRf, *R* is the wingspan length, *f* is the flapping frequency and ν is the motion viscosity of air. On this basis, by changing the wingspan length and chord length at the same time, the aspect ratio can be changed on the premise of maintaining the same Reynolds number, and the final parameter design of a series of wings can be obtained as shown in Table 1.

### 2.2. Kinematic Model

In this study, the movement pattern of imitation hummingbird wings is constructed based on the structure and movement pattern of hummingbird wings. The hummingbirds’ flapping process is converted into the overall deflection movement of the rigid wings so as to simplify the morphology and kinematic model of hummingbirds. As shown in Figure 2a, the study defines two coordinate axes, the global fixed coordinate *OXYZ* and the wing coordinate *O′X′Y′Z′*, where the origin coordinates *O* and *O′* are fixed at the root of the leading edge of the wing, and the wing coordinates change with the movement of the wing, *O′X′* is parallel to the normal direction of the wing surface, and *O′Y′* is always parallel to the extension direction of the wing root. *O′Z′* is always parallel to the leading edge of the wing.

For this study, the motion of a flexible hummingbird wing is simplified to obtain a rigid imitation wing whose kinematic mode can be described as a flutter motion rotating around the flapping axis and a tilt motion rotating around the leading edge of the wing. In the process of motion, there is a fixed flapping plane, which is described as the *XOZ* plane in the construction of a global fixed coordinate system, and the leading edge of the wing will always be located in the flapping plane; that is, the angle between the *O′Z′* axis and the *XOZ* plane is always 0°. As shown in Figure 2, the posture of the hummingbird’s wings in the stroke can be described by two angles: stroke angle Φ and deviation angle θ. Among these, stroke angle Φ is the angle between the *O′Z′* axis and the *OZ* axis, which is used to describe the rotation angle of the wing around the flapping axis, while deviation angle θ is defined as the angle between the *O′Y′* axis and the *OY* axis, which is used to describe the angle of the wing tilting around its leading edge. Generally speaking, an angle of attack α is usually needed to describe the kinematic mode of hummingbirds’ wings, which is used to represent the angle between the flapping plane and the flight plane [26]. Studies by Usherwood and Ellington [45] show that the effects of aspect ratio on the aerodynamic characteristics of hummingbird motion are only applicable in a certain range, and there are different influencing mechanisms at larger angles of attack. Therefore, this study adopts a fixed angle of attack α = 0° and ignores the change in angle of attack in motion models. Only changes in stroke angle and deviation angle are considered.

Figure 3 shows the variation process of stroke angle Φ and deviation angle θ in a flapping period. When stroke angle Φ reaches near the amplitude, the flutter of the wing is close to the maximum value, the deviation angle θ is close to 0° and the wing reverses. When the stroke angle Φ is close to 0°, the flutter speed is the maximum, while the deviation angle θ is maintained in a larger angle, providing the main source of lift in the stroke. Equation (Equation 1) represents the governing equation of wing motion as
(1)$$\omega_{x}=\frac{d\theta }{dt}sin\Phi ,\omega_{y}=\frac{d\Phi }{dt},\omega_{z}=\frac{d\theta }{dt}cos\Phi$$
where $\omega_{x}$, $\omega_{y}$ and $\omega_{z}$ represent the angular velocity of the wing on the *X*, *Y* and *Z* axes. The study defines the angular velocities in three directions, obtaining a quasi-steady kinematic model of the hummingbird wing. By paying attention to the translational and rotational motions in the hover process in a simplified way, the study can focus on the unsteady aerodynamic characteristics of the flapping process.

### 2.3. Numerical Setup

This study uses an academic version of ANSYS-FLUENT to carry out the numerical simulation of biomimetic hummingbird wings. In this study, overset-grid function is adopted to simulate the movement of components, and user-defined functions (UDFs) were imported to control the kinematics mode of the moving region.

The governing fluid equation solved in this paper is the incompressible Navier–Stokes equations:(2)$$\frac{\partial u_{i}}{\partial x_{i}}=0$$
(3)∂ui∂t+∂uiuj∂xj=−∂p∂xi+1Re∂2ui∂xi∂xj

Since the wing model is always at a low reference speed during the flapping phase, the density varies little with pressure at lower flight speeds, and the gas can be regarded as an incompressible fluid. In Equation (Equation 3), $u_{i}$ is the velocity component, *p* is the pressure component, and Re is the Reynolds number.

The flow field grid generated by the overset method is shown in Figure 4. The flow field is divided into three parts: the internal domain, external domain and wing region. The wing model is placed in the wing region, where the mesh overlaps with the internal domain. Triangular surface elements and tetrahedral elements are used in the wing region, which is convenient for the movement of the area. The internal and external domain fill the entire computing domain, and all these domains use structural grid to improve the calculation accuracy. The number of grids is 1.03 million in the wing region, 1.11 million in the internal domain, and 1.38 million in the external domain; thus, the total number of grids is 3.52 million. While the wing goes through its kinematic period, the wing moves together with the wing region throughout the whole period and is always inside the inner domain.

### 2.4. Verification and Validation

To determine the reasonableness of the numerical methods, verification and validation are required. The verification of the computational model can be divided into four parts: mesh and grid independence verification, time step independence verification, turbulence model verification and comparison with the experimental results of an existing FMAV.

By modifying the size of the cryptographic mesh in the inner domain, the study constructed three different numerical domain grids with 1.4 million, 1.8 million and 2.4 million cells respectively, and the lift coefficient of each mesh structure was evaluated. The results are shown in Figure 5a. It can be seen that the overall variation trend of the three grid numbers in the flapping period is almost the same. Therefore, the grid we have adopted is perfectly adequate. On this basis, three different time steps, $\Delta t=2\times 10^{-5}$ s, $1\times 10^{-5}$ s and $5\times 10^{-6}$ s, were introduced to calculate flapping period during flight. The results are shown in Figure 5b. The values of lift coefficients under these three time steps are roughly consistent, which means that time steps in this range have little influence on the calculated results.

Since flapping wing flight is a typical turbulence problem at moderate Reynolds number, the k-ϵ renormalization group (RNG) model was used for calculation. The k-ϵ is a model widely used in turbulence problems, and the RNG theory provides an analytical formula to consider the viscosity of flow at low Reynolds numbers, which has higher reliability and accuracy over a wider range of Reynolds numbers. In order to verify the rationality of turbulence model, the k-ϵ RNG model and the large eddy simulation (LES) method were used to numerically simulate the flapping lift coefficients of the same aspect ratio wing model. The results are shown in Figure 6. It can be seen that the lift curves calculated by the two methods are very similar, and k-ϵ RNG model has a much faster calculation speed than LES method, so it is reasonable to choose the k-ϵ RNG turbulence model for calculation.

Finally, the experimental results are compared with those of an actual FMAV to verify the accuracy of the numerical method. The insect-like FMAV developed by Harbin Institute of Technology (HIT) is selected as a comparison object [46]. By changing the output frequency of the motor, HIT’s FMAV changes the common frequency of the wings, and the lift force under different flapping frequencies is obtained. It should be noted that in HIT’s experiment, when the flapping frequency of the prototype exceeds 14 Hz, the instantaneous torque is too large, resulting in data errors. The study adopted a similar wing model to simulate its motion mode and obtained the calculation results as shown in Figure 7. It can be seen that, excluding the sensor problems existing in the experiment at a larger frequency, there is little difference between the numerical calculation and the experimental results at a lower flapping frequency below 14 Hz, so our calculation results can be considered precise and accurate.

## 3. Results and Discussion

By simulating the flapping process of hovering hummingbirds, the flight characteristics of flapping wings with different aspect ratios are obtained. The lift drag characteristics of this flight are first reported, followed by an analysis of the wing aerodynamic characteristics and flight efficiency, resulting in an aspect ratio design with good flight performance. Then, the three-dimensional flow field structure is observed in order to simulate the flow of air when a hummingbird flaps its wings. Following this, the influence of wing aspect ratio on hovering motion is evaluated.

### 3.1. Aerodynamics of Flapping Wing

When the hummingbird hovers, there is no incoming velocity; thus, it can be concluded that the appropriate direction of the lift vector is perpendicular to the velocity direction of the wing stroke itself and aerodynamic torque owing to the drag being parallel to the stroke velocity [26]. The lift and drag coefficient of a hovering hummingbird wing in the flapping process are both described in two different ways according to the study of Shahzad [19] and Kryut [26]. In Shahzad’s study, the lift and drag coefficients are defined as in Equation (Equation 4):(4)$$C_{L(Sh)}=\frac{2F_{L}}{\rho_{f}U^{2}S},\;C_{D(Sh)}=\frac{2F_{D}}{\rho_{f}U^{2}S}$$
where $\rho_{f}$ is the fluid density, and *U* the reference velocity ($U=\sqrt{S}\times f$). In our study, in order to maintain a constant Re number, the wing chord length and wingspan change simultaneously, as shown in Table 1, so that the wing area *S* remains unchanged with different wings. Thus, for wing models with different AR, the reference velocity *U* remains unchanged. In the formula corresponding to $C_{L(Sh)}$ and $C_{D(Sh)}$ in Equation (Equation 4), the constant *S* and *U* means that the magnitude of lift and drag force has the same variation trend as the corresponding lift and drag coefficient.

In Equation (Equation 4), the characteristic velocity is simply expressed as a function of the wing area, which allows different wing configurations to have exactly the same characteristic velocity. Meanwhile, the assumption of Equation (Equation 4) regards the wing as a uniform plate, while in fact the air flow is not evenly distributed along the surface of the wing. The study further introduces a method similar to Kryut’s to calculate lift and drag coefficients:(5)$$C_{L(Kr)}=\frac{2F_{L}}{\rho {\left(2\pi f\right)}^{2}R_{2}^{2}S},\;C_{D(Kr)}=\frac{2F_{D}}{\rho {\left(2\pi f\right)}^{2}R_{3}^{3}S}$$

In Equation (Equation 5), the airflow velocity gradient distributed along the surface of the wing is considered, and the forces on different parts of the surface are not uniform. Using the average lift and drag force to replace lift and drag coefficient requires special processing with $R_{2}$ and $R_{3}$ [8], where $R_{2}$ and $R_{3}$ are the second and third moments of the area respectively, representing the velocity gradient along the wingspan and is expressed as
(6)$$R_{2}=\sqrt{\frac{1}{S}\int_{0}^{R}r^{2}c\left(r\right)dr},\;R_{3}=\sqrt[{3}]{\frac{1}{S}\int_{0}^{R}r^{3}c\left(r\right)dr}$$
where dr is the infinitesimal wingspan, and c(r) is the local wing chord length corresponding to the location of wingspan. The local blade element area integral is used to obtain the second and third moments of the area. By introducing $R_{2}$ and $R_{3}$ into the average lift and drag force formula, the wing is divided into infinite thin blade elements along the wing span, and the force acting on each element is integrated to obtain the final lift force and drag coefficient.

The aerodynamic characteristics of five different wing models of hovering hummingbirds are investigated by numerical simulation. The wing models are designed to have different aspect ratios, ranging from 2 to 6, covering the real wing shape (from 3 to 4.5) of natural hummingbirds. The results are shown in the following figures.

In Figure 8a, it can be seen in detail how the lift coefficient of the wing changes with time in two complete flapping periods. Similarly to other flapping creatures such as birds and insects, the flapping movement pattern of hummingbirds can also be divided into upstroke and downstroke. Different from birds flapping their wings up and down, however, hummingbirds flapping their wings in a nearly horizontal plane, and the angle of attack (the angle between the flapping plane and the horizontal plane) varies in a range of close to 0°. It is approximately considered that the flapping plane of hummingbird wings is located in the horizontal plane. This unique flapping mode of a hummingbird wing enables it to provide greater lift. As can be seen from Figure 8a, a hummingbird wing provides positive lift for most of the time in the whole stroke phase, thus generating a greater lift for hover during the whole flapping period. The $C_{L(Sh)}$ defined by Shahzad’s method is on the order of more than 100, much larger than the usual value, but the results are still instructive. Meanwhile, $C_{L(Kr)}$ changes over time in much the same way, but its value is closer to the usual lift coefficient value. However, during the first half of $C_{L(Kr)}$ increasing in each stroke, the rate of increase varies greatly. In Figure 8b, the wing with a smaller AR has a higher slope of its lift line. Similarly, in midstroke, the wing with a smaller AR has a higher peak of its lift curve and a faster speed to reach the peak of its lift, and has a higher lift coefficient.

These features lead to the results in Figure 9. The average lift coefficient ${\bar{C}}_{L(Sh)}$ increases with the increase in AR, showing a roughly linear relationship. As mentioned before, the variation trend of lift coefficient is the same as that of lift force, and it can be concluded that under the premise of maintaining constant Re number and wing area, the wings with larger AR have higher total lift force during flapping. On the contrary, it can be seen that ${\bar{C}}_{L(Kr)}$ decreases with the increase in AR, but the overall rate of decline is not constant. When AR < 4, the slope of the lift line continuously decreases, which means the decreasing trend of the lift coefficient continuously slows down. When AR = 4, the lift coefficient remains almost constant. When AR > 4, the slope of the lift line keeps increasing, and the reduction speed of the lift coefficient gradually accelerates.

This is followed by the curve of the drag coefficient. At the end of each stroke phase, the stroke angle of the flapping wing reaches the maximum and the direction of the stroke velocity reverses. The direction of drag force is defined as parallel to stroke velocity such that at the end of each stroke phase, the direction of drag force also reverses. In Figure 10, it can be seen that the sudden change in drag force coefficient caused by the reverse of drag force. In each stroke, the drag increases first and then decreases. Near the end of each stoke, the drag coefficient drops to 0, which means that when the flapping direction is about to reverse, and the flapping generates air flow to provide a positive pushing effect for the movement of the wing. All these wings have a similar growth and decline trend, but in the first half, wings with higher AR have much faster increases. According to Figure 11, similar to ${\bar{C}}_{L(Sh)}$, ${\bar{C}}_{D(Sh)}$ shows a change process of continuous increase when AR becomes larger, which means that when the AR of the wing grows larger, it will encounter higher drag force during flapping. On the contrary, the ${\bar{C}}_{D(Kr)}$ curve also shows a change process of continuous decrease with the increase in AR. However, in the ${\bar{C}}_{D(Kr)}$ curve, the slope change is not obvious, which is closer to linear change than the ${\bar{C}}_{D(Kr)}$ curve.

### 3.2. Flight Performance of Flapping Wing

In general, the lift coefficients and drag coefficients show completely different trends through different treatment methods. Therefore, the lift–drag ratio is introduced as a reference to more effectively analyze the differences in aerodynamic characteristics of different AR wings. Lift–drag ratio is an important parameter to evaluate the aerodynamic efficiency of aircraft. It is defined as the ratio of lift coefficient to drag coefficient.

Figure 12 clearly shows the difference in average lift–drag ratio of hummingbird wings with different ARs during two flapping periods. Compared with Figure 9 and Figure 11, although the two calculation methods are completely different in the variation trend of lift and drag coefficients, their lift–drag ratio results show great similarity, which demonstrates that lift–drag ratio is a more appropriate parameter for evaluating aerodynamic characteristics in this study. In Figure 12, the curve has an obvious upward trend when AR is low, and the lift–drag ratio increases obviously with the increase in AR, which is also consistent with some understandings of fixed wing aircraft. When AR becomes larger, the growth rate of the curve slows down, and a relatively stable lift–drag ratio is maintained. However, when AR is equal to about 4, the increase rate of lift–drag ratio accelerates sharply and reaches a peak. In this range, the lift–drag ratio characteristics of the wing are even higher than those of the higher aspect ratio, and it has more excellent aerodynamic characteristics.

During hovering flapping, drag is defined as the force in the direction opposite to the stroke velocity of the wing and is a major factor affecting the power consumption of the flapping process. In the study of Kryut [26], in order to comprehensively evaluate the influence of aspect ratio on hover performance at low Reynolds number, power factor (PF) was introduced for comparison [47]:(7)$$PF=\sqrt{\frac{{C_{L}}^{3}}{{C_{D}}^{2}}}$$

The PF is a parameter directly proportional to aerodynamic power and is an indicator of aerodynamic efficiency performance of a certain wing, as it measures how much weight can be lifted per unit of aerodynamic power. Since power is proportional to the reciprocal of PF, the higher the power factor, the lower the required aerodynamic power.

Figure 13 shows the power factor curves of hummingbird wings with different aspect ratios in the hovering period. Different from the change trend of lift–drag ratio curves, the power factor curve shows an obvious decline with the increase in wing AR, which means that the aerodynamic efficiency and energy efficiency of the wing with high AR are lower in flouncing flight. This also explains why most flapping creatures in nature have smaller AR wings (a few creatures with long AR wings, such as dragonflies, have a lower equivalent AR considering their unique structure with two pairs of wings). However, when AR ≈ 4, the power factor curve increases instead of decreasing, reaching an extremely significant peak, which means that the flapping wing with AR ≈ 4 has better aerodynamic efficiency under this motion mode. A similar situation also appears in Figure 12, demonstrating that the flapping wing has better aerodynamic characteristics and aerodynamic efficiency near this AR, which is also consistent with the actual AR (3–4.5 [41]) of hummingbirds in nature.

### 3.3. Flow Field Analysis and Discussions

This study uses two different methods to obtain lift and drag coefficients of biomimetic hummingbird wings in the hovering process, and a large difference in the results is obtained. This difference is closely related to the earlier calculation formula. In Equation (Equation 4), the characteristic velocity remains unchanged with different AR wings, and the denominator in the formula is always constant, so that there is a linear relationship between coefficients and forces. This means that the trends in $C_{L(Sh)}$ and $C_{D(Sh)}$ can be used to better evaluate the total lift and drag force produced on the wing. On the other hand, in Equation (Equation 5), the characteristic velocity used to calculate the lift coefficient and drag coefficient is a characteristic product obtained by integrating along the wingspan direction, which changes with the wing design and is closely related to the wingspan difference. The variation trend of $C_{L(Kr)}$ and $C_{D(Kr)}$ obtained can thus better show the differences caused by the uneven distribution of aerodynamic characteristics along the wing surface and can allow more comprehensively evaluating the aerodynamic efficiency of the wing.

From the time history curves of lift coefficient and drag coefficient, it can be seen that the lift and drag coefficient of biomimetic hummingbird wings with different AR reach their peak at *t* = 0.25 T. It is also at this moment that the lift coefficient and drag coefficient of biomimetic hummingbird wings with different AR reach their peak, and the difference in the aerodynamic characteristics between different AR wings reaches its maximum. Therefore, three types of wings with AR = 3, 4 and 5 are selected to explore the flow field characteristics at *t* = 0.25 T. Starting from the pressure nephogram and vorticity diagram around the wings, the study expected to investigate the reasons for the differences in lift and drag characteristics of different AR wings and conduct a mechanical analysis of the influence of AR on the aerodynamic characteristics of the wings.

Figure 14 shows the pressure distribution on the upper and lower surfaces of different AR wings. On the upper surface of the wing, there is a relatively obvious low pressure area. The low pressure area is roughly located at the leading edge of the wing and concentrated near the half of the wingspan. From the leading edge of the wing to the tail edge, and from the half of the wing to wing tip and wing root, the effect of the low pressure area weakens. The pressure on the upper surface gradually returns to normal pressure. On the lower surface of the wing, there is a high pressure area located at the position of the leading edge of the wing near the wing tip. As the high pressure area approaches the wing root, its pressure gradually lowers. The low pressure area on the upper surface and the high pressure area on the lower surface act together to produce a pressure difference on the wing surface. The direction of the pressure difference is from the lower surface to the upper surface, which is the most important source of lift and drag force on the wing during the flapping process.

In Figure 14, it can roughly be observed that the distribution range and pressure of the low pressure area on the upper surface and the high pressure area on the lower surface. In order to study the pressure difference of the wing in more detail, the study intercepts the flow field at the wing tip and 50% semi-span location along the vertical direction of the wing surface to observe the pressure contour on the section. Figure 15a–c is the flow field intercepted at the wing tip to observe the high pressure area located below the wing. By comparing the pressure contour of the three different AR wings, it can be seen that with the increase in AR, the radius and central pressure of the high pressure zone are also increasing such that the lift force on the wing is also increasing. Further, the flow field was intercepted at the semi-span location to observe the characteristics of the low pressure region. With the growth in AR, the low pressure region shows a similar change trend. When the pressure continues to decrease, the range of the low pressure region also increases such that the lift and drag force on the wing increase with the growth in AR. This also explains the variation trend of $C_{L(Sh)}$ and $C_{D(Sh)}$ with AR in Figure 9 and Figure 11.

By further observing Figure 15d–f, it can be noted that with the growth in AR, while the range of the low pressure region keeps increasing, the height of its center position also decreases, gradually approaching from the leading edge of the wing to the tail edge. Shifting the position of the low pressure area changes the direction of the resultant force on the wing due to the pressure difference between the upper and lower surfaces. As the height of the low pressure area decreases, so does the angle between the resultant force and the horizontal plane, meaning that the resultant force exerts more force in the horizontal direction, that is, the direction of drag force. This effect causes lift and drag to grow at different rates. As reflected in the lift–drag ratio, it can be seen that the lift–drag ratio reaches a peak when AR = 4 but decreases when AR > 4, which means that when AR further increases, the growth rate of drag will exceed that of lift. Although both lift and drag are increasing, their aerodynamic characteristics will not be monotonically improved.

The vortices on the section at 50% wingspan are calculated based on Q criterion, and the obtained vortices diagram is shown in Figure 16. It can be seen that the leading edge vortex (LEV) extends from the leading edge of the wing, and the LEV develops to the rear along the wing surface and remains above the wing. Combined with Figure 15, it can be seen that the LEV at the semi-span position is the main source of the low pressure area there, that is, the main factor providing the lift and drag of the wing at this moment. With the continuous growth in AR, LEV develops more completely, and more significant LEV above the wing causes the wing to suffer greater lift and drag, thus causing $C_{L(Sh)}$ and $C_{D(Sh)}$ to increase with the growth in AR.

In Figure 16, in addition to LEVs, a trailing edge vortex (TEV) extending from the trailing edge of the wing can be seen. These TEVs are not fully developed at semi-span and have little impact on the aerodynamic characteristics. However, the TEV continues to develop in the direction of wingspan along the tail edge of the wing. In Figure 17, at 75% wingspan position, it can be seen that the vortices are developed to a great extent as TEV approaches the wing tip, and the effect becomes increasingly significant. In wings with smaller AR, TEV development is not complete because of the smaller wingspan, and the influence is less significant. This trend is less evident in smaller AR wings (AR = 3), where changes in vorticity brought on by TEV are clear from AR = 4. When AR continues to increase to AR = 5, the larger aspect ratio enables TEV to have more sufficient distance for development, and the fully developed TEV has begun to affect the LEV. In Figure 17c, compared with the other two AR wings, the LEV has begun to shed. It does not stay over the wing like it does at semi-span, as LEV acts as the main source of lift provided in hovering process. If TEV development adversely affects LEV, the lift force of the wing will decrease faster, and its aerodynamic characteristics will be worse at a farther position along wingspan. This also explains the variation trend of $C_{L(Kr)}$ and $C_{D(Kr)}$. The lift and drag coefficients obtained in this way take into account the uneven after-weighted average of the spanwise development of the wing surface. The influence of TEV on LEV becomes more significant with the increase in AR, which causes faster lift and drag drop outward along the wingspan. Thus, $C_{L(Kr)}$ and $C_{D(Kr)}$ decrease with the increase in AR.

As the AR of the wing increases, TEV develops more completely, resulting in LEV shedding. However, in Figure 17, at 75% wingspan, the TEV of the wing with AR = 4 has developed significantly, but its LEV is still attached to the leading edge of the wing and does not shed. Continuing toward the wing tip, after reaching 80% wingspan, it can be seen in Figure 18b that more significant TEV eventually leads to LEV shedding, but at the same location, in Figure 18a, signs of LEV shedding also begin to occur. This means that TEV is not the only factor contributing to LEV shedding. Other factors such as inertia force and induced drag will also cause LEV shedding near the wing tip. LEV shedding appears significantly earlier when AR = 5, and there is no significant difference in the shedding position of LEV compared with AR = 3 and AR = 4. LEV shedding has less adverse effect on AR = 4 wings, which makes AR = 4 wing lift–drag ratio and power factor more excellent, with better aerodynamic characteristics and aerodynamic efficiency.

## 4. Conclusions

This study investigated the effects of aspect ratio on hummingbird biomimetic wings. Through the CFD method, the hovering and flapping process of the hummingbird wing model was numerically simulated, and the lift and drag characteristics of different AR wings were quantitatively analyzed. After studying the flow field characteristics around the wing, the difference and mechanism of the aerodynamic characteristics of different AR airfoils are explained through pressure nephograms and vortices diagrams. The results show that as the AR increases, the lift and drag of biomimetic hummingbird wings also increase during hovering and flapping. This is because with the increase in AR, the vortices generated on the wing can develop more completely along wingspan, and their characteristics become more pronounced, resulting in a greater pressure difference between the upper and lower surfaces of the wing, which increases the lift and drag force. On the other hand, the aerodynamic characteristics of the biomimetic hummingbird wing do not monotonously improve with the growth in AR. While the leading edge vortex develops, the tail edge vortex also develops along the wingspan. As the AR increases, the increase in wingspan provides more ample distance for the TEV to more fully develop. The influence of TEV causes earlier LEV shedding, which results in the faster lift drop near the wing tip. This effect causes the lift–drag ratio and power factor to decrease with the further increase in wing AR. In general, the biomimetic hummingbird wing has better aerodynamic characteristics and higher aerodynamic efficiency in the range of AR ≈ 4, which also conforms to the actual aspect ratio of hummingbird wings in nature.

## Figures and Tables

**Figure 1 biomimetics-08-00216-f001:**
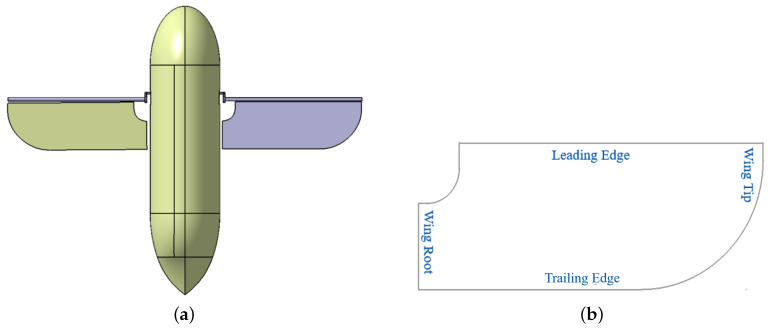
(**a**) The hummingbird-like MAV design and (**b**) the basic hummingbird-like wing mode used in this study.

**Figure 2 biomimetics-08-00216-f002:**
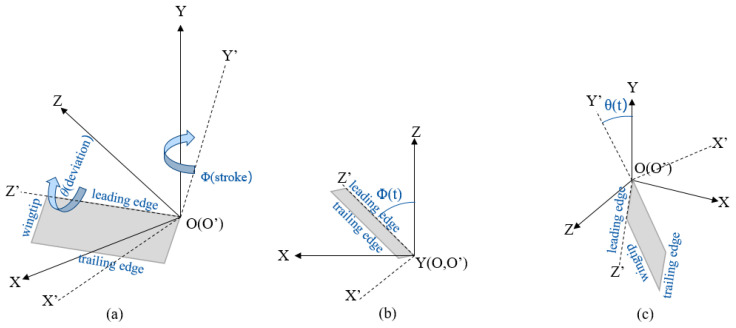
The coordinates and angles used to describe hummingbird-like wing’s kinematic model: (**a**) the global fixed coordinate *OXYZ* and the wing coordinate *O′X′Y′Z′*, and detailed kinematic model; (**b**) the definition of stroke angle Φ and (**c**) deviation angle θ.

**Figure 3 biomimetics-08-00216-f003:**
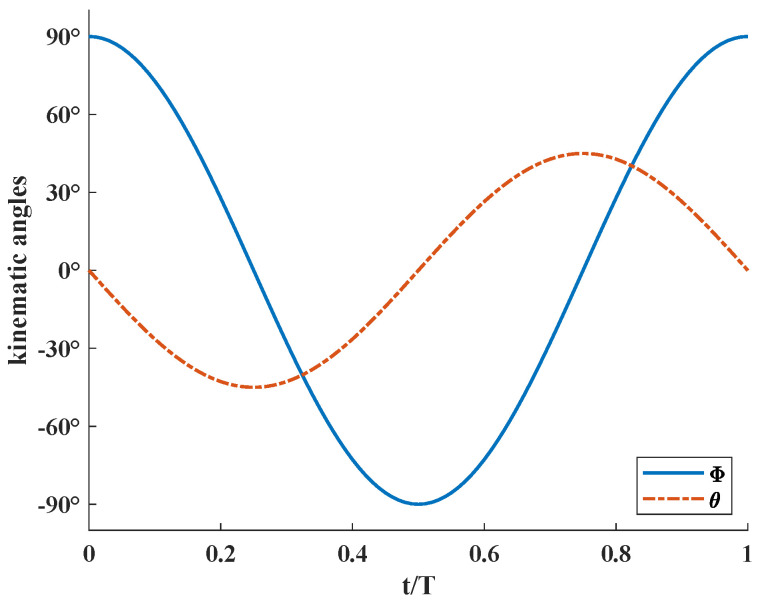
The kinematic model represented by stroke angle Φ and deviation angle θ.

**Figure 4 biomimetics-08-00216-f004:**
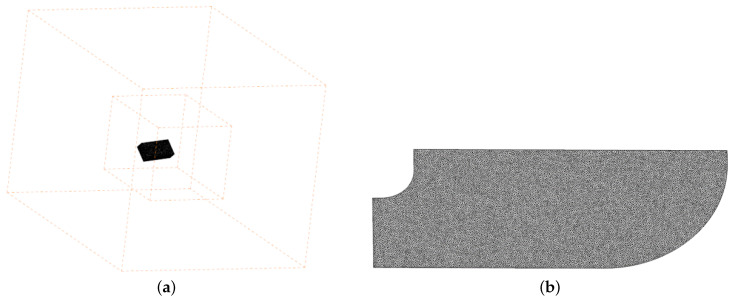
The mesh design of (**a**) the whole computation domain and (**b**) wing surface mesh.

**Figure 5 biomimetics-08-00216-f005:**
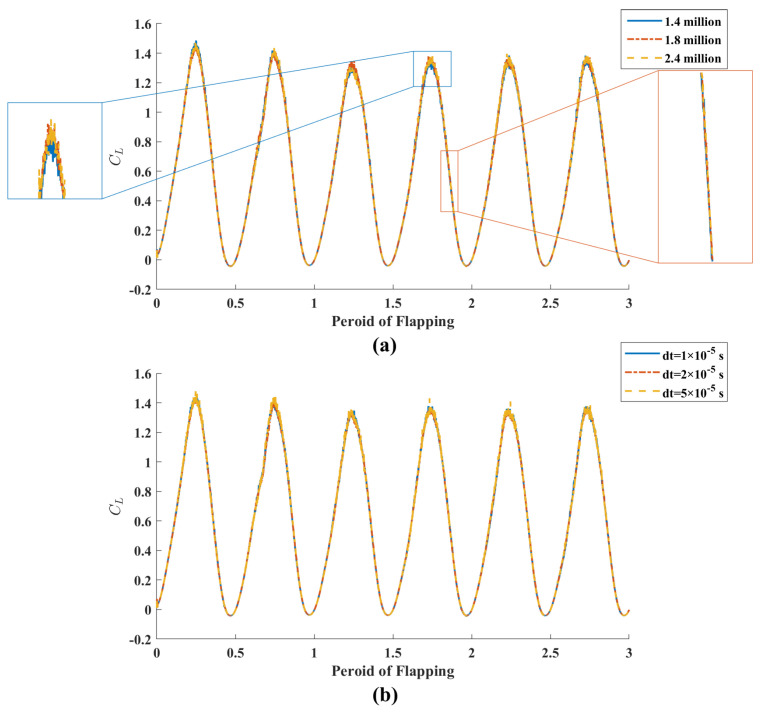
Lift coefficients of (**a**) grid independence and (**b**) time step independence verification.

**Figure 6 biomimetics-08-00216-f006:**
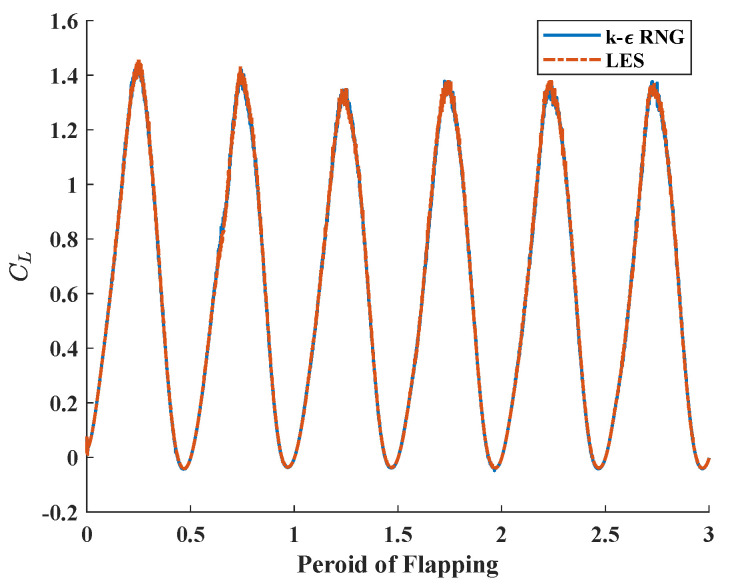
Lift coefficients derived from different computational models.

**Figure 7 biomimetics-08-00216-f007:**
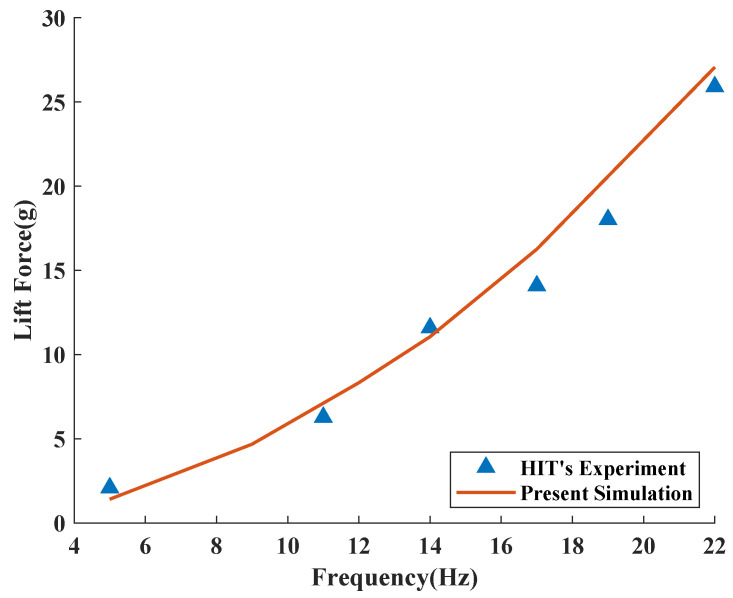
The comparison of lift coefficients between numerical calculation and Liu’s experimental results [46].

**Figure 8 biomimetics-08-00216-f008:**
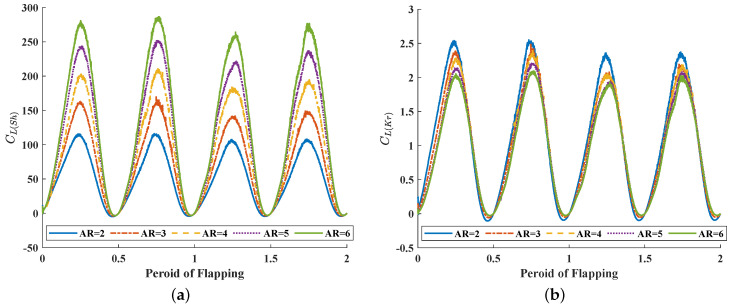
The time history of (**a**) $C_{L(Sh)}$ and (**b**) $C_{L(Kr)}$ of different AR wings.

**Figure 9 biomimetics-08-00216-f009:**
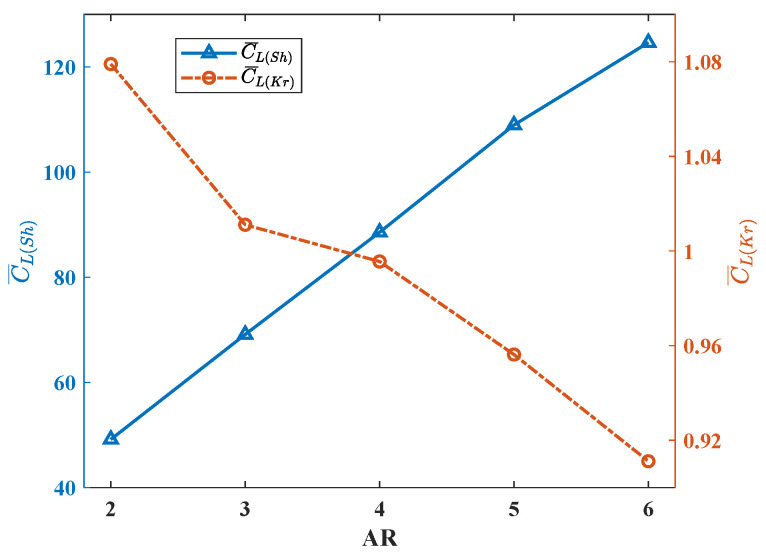
The average lift coefficients of different AR wings.

**Figure 10 biomimetics-08-00216-f010:**
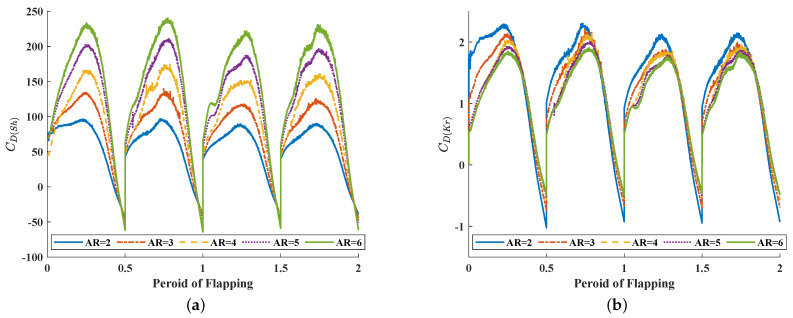
The time history of (**a**) $C_{D(Sh)}$ and (**b**) $C_{D(Kr)}$ of different AR wings.

**Figure 11 biomimetics-08-00216-f011:**
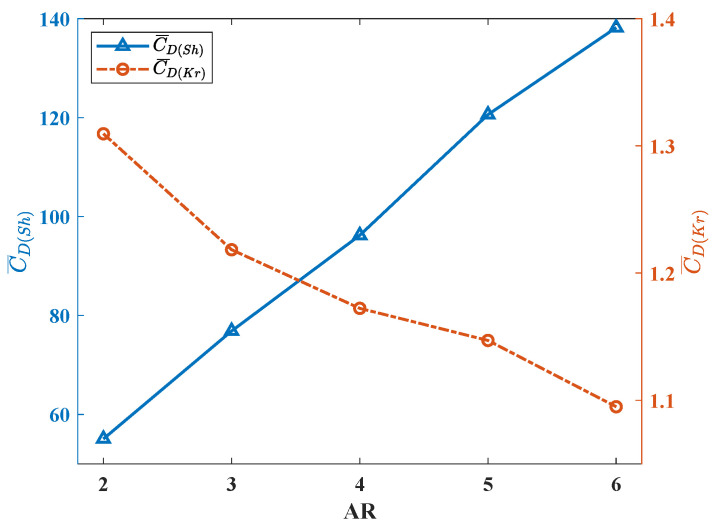
The average drag coefficients of different AR wings.

**Figure 12 biomimetics-08-00216-f012:**
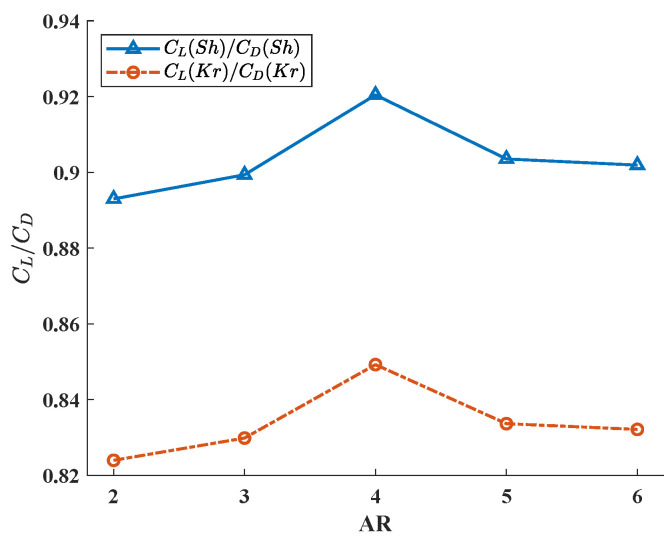
The lift–drag ratio ($C_{L}/C_{D}$) of different AR wings.

**Figure 13 biomimetics-08-00216-f013:**
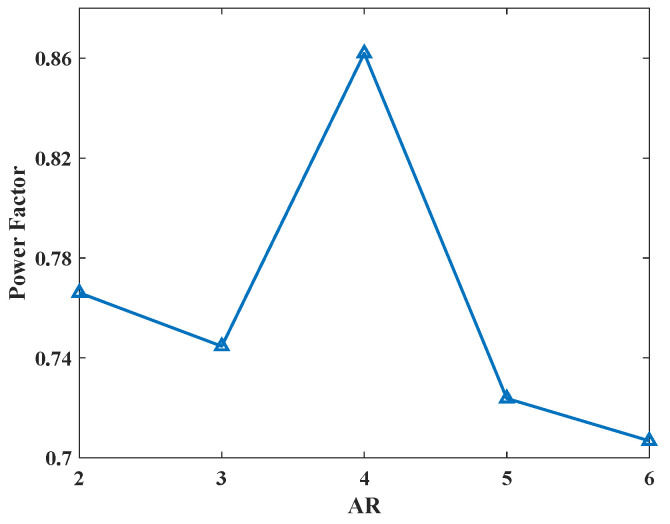
The power factor of five different AR wings.

**Figure 14 biomimetics-08-00216-f014:**
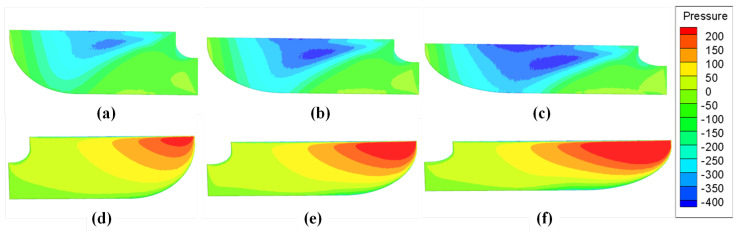
Surface pressure distribution on the upper surface of (**a**) AR = 3, (**b**) AR = 4 and (**c**) AR = 5 and on the lower surface of (**d**) AR = 3, (**e**) AR = 4 and (**f**) AR = 5.

**Figure 15 biomimetics-08-00216-f015:**
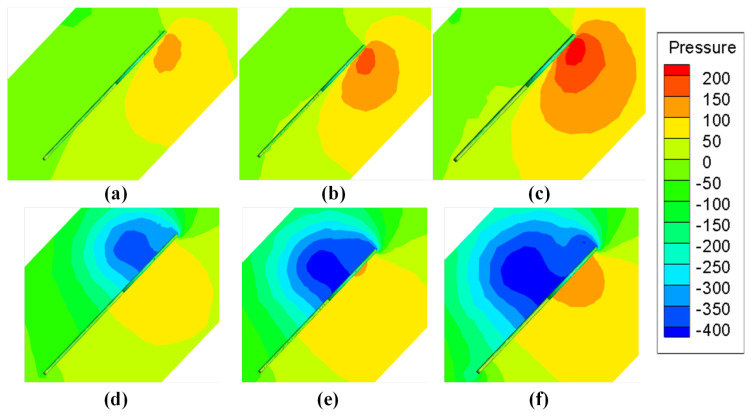
The pressure contours at wing tip of (**a**) AR = 3, (**b**) AR = 4 and (**c**) AR = 5 and at 50% semi-span location of (**d**) AR = 3, (**e**) AR = 4 and (**f**) AR = 5.

**Figure 16 biomimetics-08-00216-f016:**
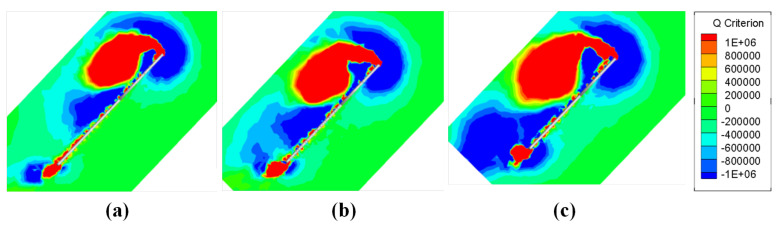
The vortices diagram at 50% semi-span location of (**a**) AR = 3, (**b**) AR = 4 and (**c**) AR = 5.

**Figure 17 biomimetics-08-00216-f017:**
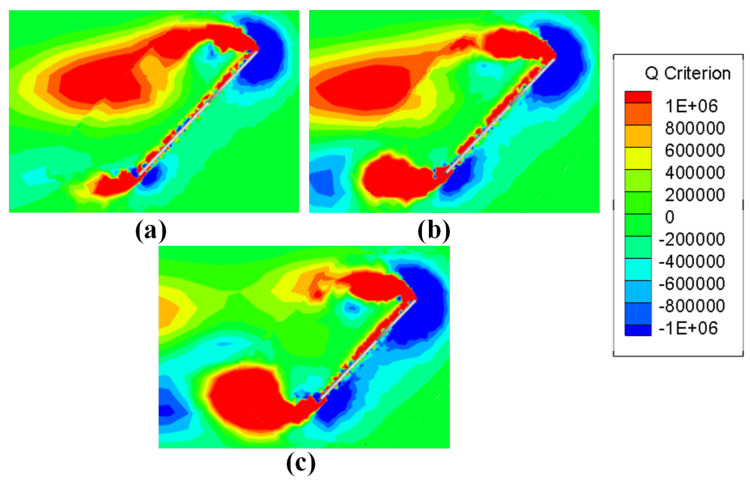
The vortices diagram at 75% semi-span location of (**a**) AR = 3, (**b**) AR = 4 and (**c**) AR = 5.

**Figure 18 biomimetics-08-00216-f018:**
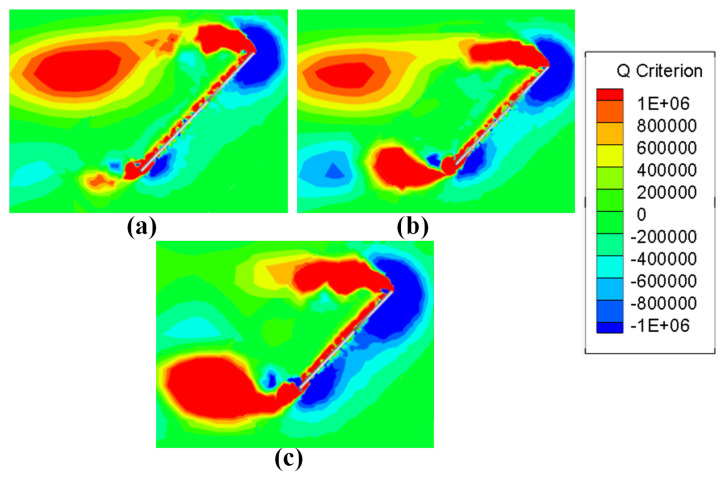
The vortices diagram at 80% semi-span location of (**a**) AR = 3, (**b**) AR = 4 and (**c**) AR = 5.

**Table 1 biomimetics-08-00216-t001:** The parameters of a series of wing models used in this study.

Wingspan (mm)	Chord Length (mm)	Flapping Frequency (Hz)	Aspect Ration	*Re*
76.0	38.0	22	2.0	27,000
93.1	31.0		3.0	
107.0	27.0		4.0	
120.2	24.0		5.0	
131.7	22.0		6.0	

## Data Availability

The data that support the findings of this study are available from the corresponding author upon reasonable request.

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
