# Peer review of "Aspect Ratio Effects on the Aerodynamic Performance of a Biomimetic Hummingbird Wing in Flapping"

_biomimetics, 2023, doi:10.3390/biomimetics8020216_

Round 1

Reviewer 1 Report

The flapping hover of a hummingbird is fascinating; still keeps researcher puzzled about how controlled and effortless it appears. The first question that comes to mind is: are these birds smarter than us? The easiest way to answer is to estimate the required power to weight ratio – which is heavily documented for man-made machines and completely ignored here (yes, I know the biologists are not aerospace engineers). Because in hover there is no forward speed, instead of using efficiency, engineers refer to ‘disk loading’ as shown below. 

With that said, it appears that the authors used a commercial code and set it up for a flapping motion – not an easy task. Assuming that FLUENT was correctly set up, presenting the large body of results in a short paper is also very difficult. So, in my opinion, if the reader of this paper can duplicate the effort described here and arrive at the same solutions – then this paper qualify as an archival publication.

Let me present some of my remarks as I read along this manuscript:

The introduction provides a comprehensive literature survey, but I’d like to see more low Reynolds number flapping publications (wings that generate lift in a separated flow environment – insects).

Reading through the Abstract and the Introduction, and later throughout the paper there is a ‘major engineering concern.’ The authors write about lift and drag, BUT the whole mechanism is driven by the root-bending-moments! (In simple words, the shoulder muscles strength is the limiting variable during the flight and hover – and it seems that the authors didn’t understand that!)

In the section ‘2. Materials and methods’: the fixed wing aspect ratio effects in attached flow (high Re) are clear: L/D increases with AR. It is less obvious with separated flow (as in the case of low Re, fast flapping). Again, in my opinion, L/D is NOT as important as root bending moments. Also, the Re number definition here is not conventional – actually it is changing along the wing span.

In section ‘2.2 Kinematic model’: If I understand correctly then the wing moves forward and then back – during the whole cycle the kinematic angle of attack is ‘kind of positive’. Would be nice to provide the wing actual angle of attack during the whole cycle (assuming the air is not moving).

In Fig 7: is the wing geometry is the same as in the experiments? Would be nice to present this in a nondimensional form.

In Fig. 9: the lift and drag coefficients for propeller are using tip speed (or ¾ chord speed). Why not do the same.

In the section about the ‘Results and discussion’: (I think) that the wing moves forward at a positive angle of attack and at the end of the stroke it flips upside down and return in the opposite direction. In order for an aerodynamicist to understand the results, the pressure distribution over the wing must be provided (like upper and lower pressure coeff. along the chord at several spanwise locations). This will allow to see when wing stall is present. Now recall that this is a serious unsteady motion – so the first unsteady term in the NS equation (some call it added mass) is responsible for the high pressures. (some researcher will call these loads circulatory and added mass – would be nice to see the separate contributions). Based on these remarks – Figs. 14 – 18 have no value, although Figs. 17, 18 suggest that the flow is heavily separated?

A more valuable presentation should include wing spanwise loading during the cycle, and the resulting wing root moments (e.g., at the bird’s shoulder). 

At the end, the energy consumed during the hover must be presented and perhaps plotted on the ‘disk loading’ diagram presented here. This will answer the question: how much better are birds in flight.

Author Response

Please find our response in the attachment.

Reviewer 2 Report

this paper presents the study on the aerodynamics of the flapping wing with the effect of aspect ratio considered. Though this study may be helpful for engineering to select a wing's shape, the main contribution is not given clearly as this topic is very old. Therefore, it can not be accepted until the novation can be clearly addressed. Some other concerns are as follows:

1) Lift coefficient is computed using two different methods, i.e. Shahzad method and Kryut's methods. Why did the authors decide to use these methods?  What are the physical differences between these two methods? 

2) The transient lift and averaged lift should be described using different symbols.

3)How to understand the trend of CL and CD versus AR shown in fig.9 and fig. 11?

4)Power factor indicates the power consumption on the basis of the same lift. For different AR cases, the lift is not the same. Therefore, the power factor is not reasonable to compare different AR cases.  Can the author really compare the power consumption of wings with the same lift produced?

Author Response

(The authors gave the same response as above.)
